# Development of the simulation-based German albuminuria screening model (S-GASM) for estimating the cost-effectiveness of albuminuria screening in Germany

**Paul Kairys**[1]*, **Thomas Frese**[1], **Paul Voigt**[1], **Johannes Horn**[2], **Matthias Girndt**[3], **Rafael Mikolajczyk**[2]

1 Institute of General Practice and Family Medicine, Medical Faculty, Martin Luther University Halle-Wittenberg, Halle/Saale, Germany, 2 Institute for Medical Epidemiology, Biometry, and Informatics (IMEBI), Medical Faculty, Martin Luther University Halle-Wittenberg, Halle/Saale, Germany, 3 Department of Internal Medicine II, Medical Faculty, Martin Luther University Halle-Wittenberg, Halle/Saale, Germany

* paul.kairys@student.uni-halle.de

## Abstract

### Background

Chronic kidney disease is often asymptomatic in its early stages but constitutes a severe burden for patients and causes major healthcare systems costs worldwide. While models for assessing the cost-effectiveness of screening were proposed in the past, they often presented only a limited view. This study aimed to develop a simulation-based German Albuminuria Screening Model (S-GASM) and present some initial applications.

### Methods

The model consists of an individual-based simulation of disease progression, considering age, gender, body mass index, systolic blood pressure, diabetes, albuminuria, glomerular filtration rate, and quality of life, furthermore, costs of testing, therapy, and renal replacement therapy with parameters based on published evidence. Selected screening scenarios were compared in a cost-effectiveness analysis.

### Results

Compared to no testing, a simulation of 10 million individuals with a current age distribution of the adult German population and a follow-up until death or the age of 90 shows that a testing of all individuals with diabetes every two years leads to a reduction of the lifetime prevalence of renal replacement therapy from 2.5% to 2.3%. The undiscounted costs of this intervention would be 1164.10 € / QALY (quality-adjusted life year). Considering saved costs for renal replacement therapy, the overall undiscounted costs would be—12581.95 € / QALY. Testing all individuals with diabetes or hypertension and screening the general population reduced the lifetime prevalence even further (to 2.2% and 1.8%, respectively). Both scenarios were cost-saving (undiscounted, - 7127.10 €/QALY and—5439.23 €/QALY).

**Data Availability Statement:** All files used for simulation are available from Open Science Framework (DOI 10.17605/OSF.IO/3JAQX).

**Funding:** The authors received no specific funding for this work.

**Competing interests:** The authors have declared that no competing interests exist.

## Conclusions

The S-GASM can be used for the comparison of various albuminuria testing strategies. The exemplary analysis demonstrates cost savings through albuminuria testing for individuals with diabetes, diabetes or hypertension, and for population-wide screening.

## Introduction

Chronic kidney disease (CKD) is a globally increasing challenge for health policy and represents a significant cost factor for healthcare systems [1, 2]. In 2015, CKD caused an expenditure of € 1.689 billion in Germany [3]. CKD is also directly relevant for patient-centered outcomes such as life expectancy and quality of life [4–6].

Prevention, early detection, and delaying CKD progression are of particular importance to prolong health, save lives, and lower costs [7]. Albuminuria has been shown a suitable trigger for therapeutic options [8], and blocking the renin-angiotensin-aldosterone system has been proven effective for delaying CKD progression [9]. Despite this therapeutic option, there are only a few nationwide programs for the early detection of CKD. One of the few considered cost-effective is the albuminuria testing program in South Korea [10].

In Germany, there are inconsistent recommendations. The national care guideline for type 2 diabetes recommends testing individuals with diabetes for albuminuria [11]. However, it only specifies to test individuals on diagnosis and not when or whether testing should be repeated. In contrast, a guideline for general practitioners even suggests not to test for albuminuria as the benefits of this intervention are not considered as proven enough [12]. The existing disease management program for diabetes in Germany instructs to test for albuminuria once per year but advises to start five years after the diagnosis of diabetes [13]. As these guidelines contradict each other in part, it seems necessary to devote renewed attention to this topic.

With the absence of direct empirical data and considering the challenge of an eventual necessary long-term follow-up, simulation models were applied to assess the cost-effectiveness of albuminuria screening programs. Most of the previously published models tried to guide reimbursement decisions by modeling risk groups only or used models with a small number of conditions [14]. In the management of CKD, simulation models for cost-effectiveness in Germany have only considered risk groups [15–21], and most of them did not integrate specific testing actions [18–21] nor a specific therapy [17, 20].

Using parameters from previous studies and models, we developed the simulation-based German albuminuria screening model (S-GASM) to assess the effects of albuminuria-testing in the general population in Germany.

## Methods

### Target population and time horizon

The S-GASM generates and analyzes individuals of the adult German population according to the population living in Germany in 2016 and, in a cohort framework, follows them in yearly steps until the age of 90 or until death, whichever comes first.

### Setting, location, and study perspective

Costs of this public health intervention are evaluated from the perspective of the German health care system. They include costs for physician consultation, testing, subsequent therapy, and costs of renal replacement therapy.

## Comparators

For the analyzes selected as examples, we compared the base-case population with no additional testing for albuminuria with three different testing scenarios. In the first scenario, all individuals with diabetes are tested. In the second scenario, all individuals with diabetes or hypertension are tested. The third scenario implements the screening of the entire population. In all scenarios, testing is performed every two years if the individuals did not already receive Angiotensin-converting-enzyme inhibitors (ACEI) or Angiotensin-II-receptor blockers (ARB).

## Discount rate

We show undiscounted and discounted costs and benefits for comparison. For discounting, we use a yearly discount rate of 3.5 percent for costs and benefits according to the NICE guidelines [22].

## Health outcomes

To measure the benefits of the intervention, we compare the lifetime prevalence and the average age at the start of renal replacement therapy (RRT), the average age at death, the proportion of individuals censored at the age of 90, and the average quality-adjusted life-years (QALY) per individual from the beginning of the simulation. We also compare the rate of false-positive results of the intervention.

## Measurement of effectiveness and preference-based outcomes

The effects of the intervention used in the S-GASM are based on the reduction in annual GFR loss in individuals with proteinuria when taking ACEI published by Agodoa et al. [23] and on the effects of ACEI on mortality and renal outcomes published by Strippoli et al. [24]. Agodoa et al. published an analysis of a randomized, double-blind, 3x2 factorial trial conducted in 1094 African Americans aged 18 to 70 with hypertensive renal disease. They compared the effects of ramipril, amlodipine, and metoprolol on hypertensive renal disease progression. We use these data since it is the only source of numerical effects of an ACEI on the annual GFR loss. Previously published models are also based on Agodoa et al. [25]. Stripoli et al. published a systematic review about the effects of ACEI and ARB on renal outcomes based on 43 trials and showed survival benefits for patients with diabetic nephropathy.

For estimating effects of testing for albuminuria followed by the treatment with an ACEI in a large proportion of the German adult population, we decided to simulate individuals with an initial GFR and the annual loss of GFR considering age, sex, body mass index (BMI), SBP, therapy of hypertension, Diabetes, RRT, albuminuria, testing, treatment, adherence, QALY, and survival.

As the first step, sex and age are generated based on the age and sex allocation within the German population in 2016 [26]. In order to generate a BMI of an individual, depending on age and sex, the model utilizes German BMI percentiles [27] in the Lambda-Mu-Sigma-(LMS)-format [28]. The LMS format enables the simulation of a standard deviation score (SDS) value. As individuals progress in the simulation, they keep their SDS value and remain at the initially assigned percentile, while their BMI changes according to changes in the population.

We extrapolated existing German blood pressure percentiles [29] provided in steps of 5 years to allocate the systolic blood pressure (SBP) in an age- and sex-dependent fashion similar to the BMI. To do so, we used a 3rd-grade polynomial for continuous changes and converted

them to the LMS format. For applying the SBP, the model starts by creating a normally distributed SDS value and the corresponding SBP, raising the SBP value BMI-dependent [30] and calculating the associated definitive SDS value. During model calibration, we adjusted the normal distribution of the original blood pressure values with an adjusted mean value to compensate for the right shift of the normally distributed SDS values caused by the influence of the BMI (see the S1 File for more information on calibration).

Individuals with an SBP exceeding 140 mmHg are considered hypertensive and receive treatment based on a drug distribution for antihypertensive therapy [31]. The once assigned drug does not change further in the model. Adherence to antihypertensive treatment is not modeled directly, but having no therapy is one of the options.

The model simulates the prevalence and incidence of Type 2 diabetes mellitus, using age- and sex-dependent prevalence and incidence data [32], multiplied with the relative risk for a BMI above 30 or hypertension [33]. By applying linear regression, we expanded diabetes incidence and prevalence data to obtain continuous age-depended risk changes. The information on model calibration is provided in the S1 File.

For Germany, we were not able to find suitable data for the simulation of albuminuria and GFR. Therefore, we had to rely on parameters extracted from a US-American simulation model [34], based on the National Health and Nutrition Examination Survey (NHANES). In this model, GFR loss and progression of albuminuria are continuous and not reversible. As data for the initial GFR and the prevalence of albuminuria are only valid for a simulation beginning at the age of 30, we simulated 1 million individuals with an initial age of 30 years. We derived age- and sex-dependent parameters, which we could then use for older age groups. Part of this US simulation model is an individual factor for annual GFR loss. We adopted this factor in our model but adjusted the overall GFR loss to 75% of the initial value during model calibration because the incidence of RRT, resulting from the initial data, was too high compared to published data for Germany [35] (see the S1 File).

In the S-GASM, the prevalence, incidence, and progression of albuminuria directly depend on age, sex, diabetes, and SBP. The initial GFR and the annual GFR loss directly depend on age, sex, diabetes, SBP, and albuminuria. The BMI indirectly influences albuminuria and the GFR through higher risks for hypertension and diabetes. As soon as an individual has a GFR value below 7 ml/min, an RRT is initiated [36].

For screening, we applied a two-times repeated urinary albumin-creatinine test as recommended in guidelines [7], with the sensitivity being 87 percent and the specificity being 88 percent for one test [37]. Both tests occur during the same annual follow-up, with both having to be positive to initiate therapy. For subsequent therapy, the model assigns every individual tested positive (including false-positive results) treatment with an ACEI as they are considered advantageous over ARB [9, 24]. The therapy resulting from a positive test reduces the loss of GFR [23], the progression from micro to macroalbuminuria, and the risk of death [24]. At the beginning of the treatment, individual adherence is assigned. Ninety-one percent of individuals are adherent, and the remaining nine percent are not taking the medication [38]. Non-adherent individuals will not benefit from therapy but still cause costs of medication. If individuals already receive ACEI or ARB for antihypertensive therapy, then independently of testing, they benefit from the treatment effects.

Using the official German mortality table [39] and the relative risks [4], according to GFR and albuminuria values, the model calculates the risk of death for an individual. ACEI therapy while having albuminuria reduces the risk of death [24]. We adjusted the baseline mortality on model calibration so that the overall mortality matched the official mortality table in the base-case population (see the S1 File for more information on calibration).

Using EuroQol measurement (EQ-5D) [40] based values, the model assigns a value for every year depending on the GFR [41].

## Estimating resources and costs

For calculating the costs of RRT, we used the annual costs of dialysis therapy and the associated costs (54,777 € in 2006) [42]. Due to inflation, we estimated these costs to have been at 63,000 € in 2016, as already used by earlier authors [21].

We calculated the urinary albumin-creatinine-based testing procedure costs based on the Physicians Fee Regulations 2015 [43]. Adding the individual costs of a consultation (No. 1) and a two-time measurement of albumin (No. 3735) and creatinine (No. 3585.H1), this sums up to 36.18 € per testing in 2016. We chose the middle of three available tariffs for each of the three components [43].

We estimated the costs of the ACEI at 15 € for 100 days in the first year using price comparison in German online pharmacies [44].

We compare the costs of testing and therapy costs with the differences in costs for RRT between each scenario and the base-case population.

## Currency, price date, and conversion

All costs are calculated and reported in euros and dated to the year they appear in the model.

## Choice of model

The S-GASM is designed as an individual-based microsimulation with annual follow-ups based on a Markov decision process [45]. The S-GASM was programmed in R (Version 3.5.3) [46]. This allows a modular design through the definition of functions, which are adaptable with minor code modifications. It will also enable a rapid adaptation to changes or updates of the underlying data by directly integrating external table formats.

Fig 1 gives an overview of modeled health parameters and their direct influence as considered in the model. We provide more detailed information on the model construction and all values used in the model in the S1 File.

## Assumptions

The S-GASM is a steady disease progression model of CKD based on initial GFR and the annual loss of GFR. Both are influenced by various health parameters discussed earlier. Changes in health parameters are modeled on the yearly follow-up, and disease progression is not reversible.

Upon combining risks and chances for certain health parameters of the total population with relative risks for affected subgroups, we assume the baseline risk for the non-effected group to be as low as needed for the overall risk of both groups to match the initial risk of the total population. We described the mathematical approach in the S1 File.

For modeling the initial GFR and prevalence of albuminuria, we rely on data provided by Yarnoff et al. [34]. These data contain values for individuals aged 30 and older in the US. We assumed the values for the age group of 18 to 29 to be the same as for the age of 30 due to the lack of alternative data. Upon model calibration, we also reduced the personal factor for the loss of GFR introduced by Yarnoff et al. to 75% of the initial value because of otherwise overestimating the incidence of RRT (see the S1 File for more information on calibration). A possible reason for this could be differences across the populations [47] and health care systems [48].

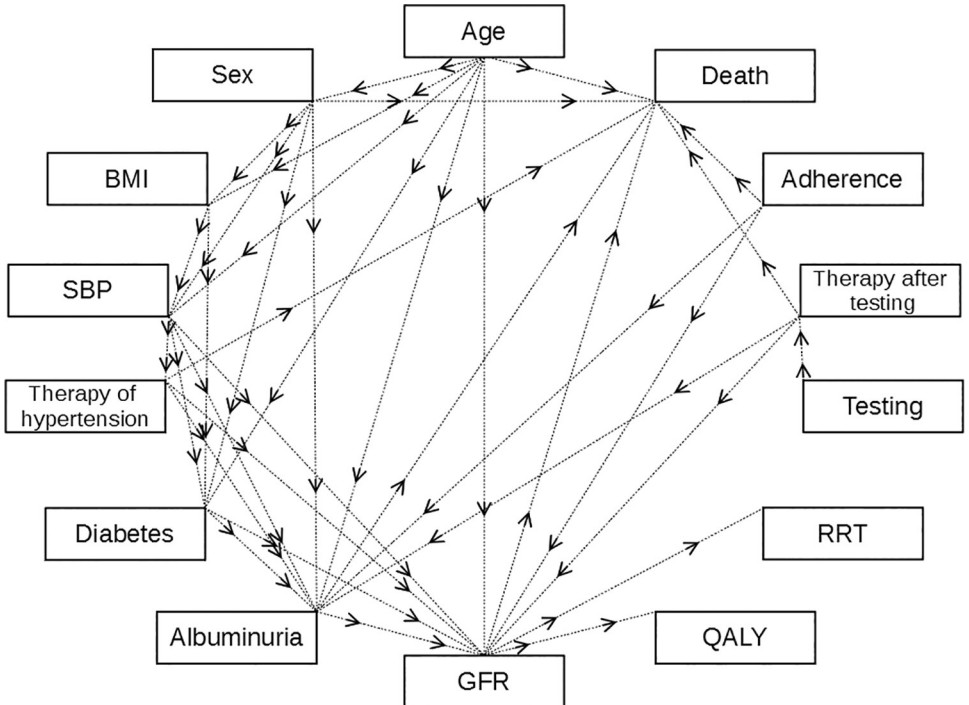

**Fig 1. Overview of modeled health parameters and direct influences.**

We decided to apply adherence only for the treatment initiated by the testing because the distribution of the therapy of hypertension [31] did not include values for adherence. This data includes individuals with no treatment, so we assume the non-adherent group to be a sub-group of the no-treatment group. We also assumed that individuals with diabetes only receive therapy with ACEI or ARB if they are also hypertensive. If an individual with diabetes is also hypertensive, the distribution for the therapy of hypertension applies. We also assume the sub-group of individuals with diabetes already receiving ACEI or ARB to contain the individuals tested for albuminuria without additional testing through our intervention.

For calculating the quality of life, we decided to simulate QALY only depending on the stage of CKD, as we were unable to find a published multivariate analysis of quality of life considering hypertension, diabetes, and CKD.

Individuals who tested positive or false-positive in testing scenarios are not tested again and cause costs for medication until their age of 90 or death.

## Analytic methods

We extrapolated the existing German BMI percentiles [27] using a 3$^{rd}$ grade polynomial for the ages from 80 to 90, as they only contain data up to 79. We also applied a 3$^{rd}$-grade polynomial to the German blood pressure percentiles [29] provided in age groups of 5 years to achieve steady progression. Additionally, we converted the extrapolated blood pressure percentiles to the LMS format [28].

We used the values for the initial GFR, annual loss of GFR, the prevalence of albuminuria, and the incidence of albuminuria published by Yarnoff et al. [34]. With values for the initial GFR and the prevalence of albuminuria available only for individuals aged 30, we simulated a cohort of 1 million individuals with that age. We followed up until the age of 90 or death to derive data for the initial GFR and prevalence of albuminuria for older age groups.

In terms of validation, we compared the distribution of age, sex, BMI, SBP, diabetes, and the risk of death of the base-case population with the published data used for model construction and present the results in the S1 File. In a simulation with 10 million individuals, we compared the simulated albuminuria prevalence, the incidence of RRT, and the time spent in the CKD stage 3a to 5 with published data [35, 49, 50]. We present and discuss the results in the S1 File.

We performed a sensitivity analysis with 1 million individuals and a gradually 5 percent increase from minus 20 to plus 20 percent in BMI, systolic blood pressure, diabetes prevalence and incidence, albuminuria incidence, and GFR loss. After that, we compared the lifetime costs of RRT per individual and the incidence of RRT and show results in the S1 File.

We present 95% confidence intervals for health outcomes and measured costs. We used the same initial simulated population in every scenario.

## Results

### Study parameters

Table 1 presents an overview of the data used in the parameterization of the model and its sources. For reasons of space, we cannot display all values directly. We present all input values in the S1 File.

### Incremental costs and outcomes

Table 2 displays health outcomes and costs of testing and therapy of albuminuria in 3 different testing scenarios compared with a base-case population with no additional testing.

**Table 1. Overview of model components and related sources of data.**

| Model Component | Sources |
|---|---|
| Age distribution | [26] |
| Sex distribution | [26] |
| BMI percentiles | [27] |
| SBP percentiles | [29] |
| SBP increase by BMI | [30] |
| Therapy of hypertension | [31] |
| Prevalence and incidence of diabetes | [51] |
| Relative risks for diabetes by BMI and SBP | [33] |
| Prevalence and incidence of albuminuria | [34] |
| Effects of ACEI on albuminuria | [24] |
| Distribution and the annual decrease of GFR | [34] |
| Effects of ACEI on GFR | [23] |
| Start of RRT | [36] |
| Sensitivity and specificity of urinary albumin-creatinine ratio | [37] |
| Adherence | [38] |
| Risk of death by age and sex | [39] |
| Relative risks for death depending on the stage of CKD | [4] |
| Effects of ACEI on the risk of death | [24] |
| Quality-adjusted life-year (QALY) | [41] |
| Costs of RRT | [42] |
| Costs of testing | [43] |
| Costs of therapy | [44] |

**Table 2. Comparison of 10 million individuals without intervention compared to 3 scenarios of testing/screening and therapy.**

| | | Base-case population | Testing all individuals with diabetes | Testing all individuals with diabetes or hypertension | Screening all individuals |
|---|---|---|---|---|---|
| Lifetime prevalence of renal replacement therapy (RRT) | | 2.549% (2.540–2.559%) | 2.318% (2.308–2.327%) | 2.190% (2.180–2.199%) | 1.844% (1.836–1.853%) |
| Average age at the beginning of RRT | | 74.141 (74.140–74.142) | 74.239 (74.238–74.240) | 74.311 (74.310–74.313) | 75.228 (75.227–75.229) |
| Average age at death | | 79.136 (79.136–79.136) | 79.162 (79.161–29.162) | 79.191 (79.190–79.191) | 79.327 (79.327–79.327) |
| Proportion of individuals censored at the age of 90 | | 27.053 (27.026–27.081) | 27.446% (27.418–27.474%) | 27.929% (27.901–27.957%) | 29.383% (29.355–29.411%) |
| undiscounted | QALYs per individual (from the start of simulation) | 30.305 (30.305–30.305) | 30.362 (30.362–30.363) | 30.433 (30.432–30.433) | 30.678 (30.678–30.678) |
| | Costs of RRT per individual | 10629.47 € (10627.69–10631.24 €) | 9975.25 € (9973.51–9976.99 €) | 9915.82 € (9514.10–9517.53 €) | 7866.54 € (7865.00–7868.08 €) |
| | Costs of testing per individual | - | 35.15 € (35.15–35.15 €) | 104.42 € (104.42–104.42 €) | 379.41 € (379.40–379.41 €) |
| | Costs of medication per individual | - | 31.75 € (31.74–31.75 €) | 99.60 € (99.59–99.61 €) | 354.29 € (354.28–354.30 €) |
| | (Costs of testing + costs of therapy) / QALY gained | - | 1164.10 € | 1598.51 € | 1966.63 € |
| | (Both costs minus saved costs of RRT) / QALY gained | - | - 10220.15 € | - 7127.10 € | - 5439.23 € |
| discounted | QALYs per individual (from the start of simulation) | 16.321 (16.320–16.321) | 16.341 (16.341–16.341) | 16.362 (16.362–16.362) | 16.434 (16.434–16.434) |
| | Costs of RRT per individual | 3988.58 € (3987.87–3989.29 €) | 3704.43 € (3703.74–3705.12 €) | 3539.75 € (3539.07–3540.43 €) | 2910.01 € (2909.40–2910.62 €) |
| | Costs of testing per individual | - | 16.85 € (16.85–16.85 €) | 53.79 € (53.79–53.79 €) | 225.37 € (225.37–22.37 €) |
| | Costs of medication per individual | - | 13.57 € (13.57–13.57 €) | 42.12 € (42.11–42.12 €) | 152.60 € (152.59–152.60 €) |
| | (Costs of testing + costs of therapy) / gained QALY | - | 1508.44 € | 2310.03 € | 3331.77 € |
| | (Costs minus saved costs of RRT) / gained QALY | - | - 12581.95 € | - 8500.65 € | - 6175.89 € |
| Individuals tested false positive | | 0 | 0.0698% (0.0682–0.0714%) | 0.1362% (0.1340–0.1385%) | 0.2810% (0.2777–0.2843%) |

Testing/screening is performed every two years in individuals without ACEI or ARB already assigned. (95% confidence intervals).

Note: population size is 10,000,000. (95% confidence intervals). Abbreviations: RRT, renal replacement therapy; QALY, quality-adjusted life-year

## Discussion

### Study findings

We present a model developed to study the effects of different testing strategies for albuminuria, based on a detailed representation of risk factors involved in the CKD progression. Based on this model, we show that introducing testing for albuminuria is potentially cost-saving when the additional costs of testing and therapy are compared with reduced costs for RRT. The cost-effectiveness applies independently from discounting. The most cost-saving scenario would be to test all individuals with diabetes every two years if they are not already taking ACEI or ARB due to other reasons. We consider the cost-effectiveness results to be slightly conservative, as the costs of testing based on the physician's fee regulation are higher than the

compensation paid for most early detection measures in Germany, and the costs for medications also apply to all non-adherent individuals.

With all scenarios of testing being potentially cost-saving, there are differences in the amount of money saved. We think that it should be a subject of future work varying the time between tests, the risk factors of the tested group, the age limits for testing, and a comparison of different testing methods before giving concrete recommendations. A corresponding matrix with four dimensions would be too extensive to be presented, evaluated, and discussed as part of this work.

## Limitations

The S-GASM as a model simplifies reality, and we made several assumptions during the modeling. Every assumption made is representing a limitation. The same applies to design and concept decisions that we made during the development of this model. Furthermore, for developing the S-GASM, it was partly necessary to use data from other countries, as for certain aspects, no data from Germany was available. The differences in population compositions and health systems could have a substantial impact [47, 48]. We tried to compensate for that by adjusting multiple components during calibration. However, these adjustments themself are limitations of this work.

Furthermore, due to combining many different study outcomes and adjusting multiple values, there is no different data set for Germany to validate the S-GASM at once as needed to establish the S-GASM as a simple method.

In addition, this model does not consider the effects on individuals who were tested false positive, such as the risk of adverse clinical effects, costs of medication, and nonclinical costs for transportation, childcare, or lost wages for additional consultations. Also, the psychological effects of being labeled at risk for chronic disease are not taken into account. Further, the number of individuals who tested false positive can influence official incidence estimates and cost-effectiveness evaluations.

We would also like to mention a problem of validating the prevalence of albuminuria in younger age groups. While the simulated values are close to the values from NHANES III [52] the model is based on and data from the PREVEND [53] study from the Netherlands are also similar, the data from Girndt et al. [49] show significantly higher values in Germany, and those by Zhou et al. [54] gives even higher values from the US. We discuss this problem in more detail in the S1 File but would like to encourage further research into the field of albuminuria in childhood and adolescence, and its effects on the individual course of albuminuria and GFR in adulthood, as this has potentially a considerable influence on our topic.

## Generalizability

The S-GASM in its current state is not entirely generalizable for other countries, but the methods used to develop the model could be adapted and adjusted. The modular design and R as a programming language allow fast and easy adjustments to updated or country-specific data for the parameters used.

## Fit with current knowledge

Our results on cost-effectiveness for testing individuals with diabetes correspond partially to the results of previous models [17–20]. However, contrary to earlier models, the S-GASM offers the possibility of an individual-based simulation of a proportion of the total adult German population. It enables to compare multiple testing conditions and to provide results for concrete testing and therapy actions.

## Supporting information

**S1 File. Model development, parameter values, sensitivity analysis, and validity analysis.**
(DOCX)

## Author Contributions

**Conceptualization:** Paul Kairys, Thomas Frese, Rafael Mikolajczyk.

**Data curation:** Paul Kairys, Paul Voigt.

**Formal analysis:** Paul Kairys, Johannes Horn, Rafael Mikolajczyk.

**Investigation:** Paul Kairys, Paul Voigt.

**Methodology:** Paul Kairys, Johannes Horn, Rafael Mikolajczyk.

**Project administration:** Thomas Frese.

**Software:** Paul Kairys, Johannes Horn, Rafael Mikolajczyk.

**Supervision:** Thomas Frese.

**Validation:** Paul Kairys, Matthias Girndt.

**Writing – original draft:** Paul Kairys.

**Writing – review & editing:** Paul Kairys, Thomas Frese, Johannes Horn, Matthias Girndt, Rafael Mikolajczyk.

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
