## [Decision Letter · Decision Letter 0]

3 Nov 2021

PONE-D-21-25470Development of the Simulation-based German albuminuria screening model (S-GASM) for estimating the cost-effectiveness of albuminuria screening in GermanyPLOS ONE

Dear Dr. Kairys,

Thank you for submitting your manuscript to PLOS ONE. After careful consideration, we feel that it has merit but does not fully meet PLOS ONE’s publication criteria as it currently stands. Therefore, we invite you to submit a revised version of the manuscript that addresses the points raised during the review process.

We look forward to receiving your revised manuscript.

Kind regards,

Tatsuo Shimosawa, M.D., Ph.D.

Academic Editor

PLOS ONE

Journal Requirements:

Additional Editor Comments:

I, as an academic editor, am so sorry not to reply you earlier because the original reviewers of your submission declined to re-evaluate your article and took a longer time to find suitable reviewers.

A specialist commented minor comments.

In addition to the reviewer's comments, I have some suggestions for you.

This article aimed to establish S-GASM method, then it is required to validate it in different data set. There would be no data set in Germany other than you used for this assumption and authors should acknowledge this limitation.

Reviewers' comments:

Reviewer's Responses to Questions

**Comments to the Author**

1. Is the manuscript technically sound, and do the data support the conclusions?

Reviewer #1: Yes

2. Has the statistical analysis been performed appropriately and rigorously? 

Reviewer #1: I Don't Know

3. Have the authors made all data underlying the findings in their manuscript fully available?

Reviewer #1: Yes

4. Is the manuscript presented in an intelligible fashion and written in standard English?

Reviewer #1: Yes

5. Review Comments to the Author

Reviewer #1: This cost-effective analysis of the estimated effects of albuminuria screening seems to be rigorously scrutinized by the authors. The responses to the previous reviewers seem to be addressed appropriately. Although there are certainly several limitations to this estimation, those limitations were addressed in the literature.

I have just a few more questions.

1) I am concerned about the large gap in the presence of albuminuria and assumption in the analysis, as shown in the validation analysis, in the younger population. (To begin with, could such prevalence of albuminuria be really true in the general young population?) And as the authors mentioned, although the testing costs may not be affected, several burdens such as psychological and cost will greatly affect this population. I feel it is important to be aware of any evidence on the prevalence of albuminuria in the young general population and its pathological significance. The relevance of this estimation analysis would differ according to the pathological and treatment related significance (with ACEI/ARBs) in this population. How do the authors relate to this? Would the evidence really suggest a conservative assumption for this model (eg if treating young people with ACEI/ARBs have no influence on future outcomes) ? Maybe to be more conservative, it could be suggested that this model be better use for those above 40 years old?

2) Line 163-166 the authors describe the factors integrated in the model but should not BMI be in there to?

6. PLOS authors have the option to publish the peer review history of their article (what does this mean?). If published, this will include your full peer review and any attached files.

Reviewer #1: **Yes: **Hiroo Kawarazaki

---

## [Author Response · Author response to Decision Letter 0]

18 Dec 2021

We uploaded a separate rebuttal letter, a marked-up copy of our manuscript and an unmarked version.

We edited our file naming to meet the style requirements.

We added a DOI for our data, including all files to simulate cohorts with different parameters, and the data we analyzed in our examplary simulations.

We reviewed our reference list and have not found a retracted paper.

We uploaded our figure to PACE to ensure it meets PLOS requirements.

1) I am concerned about the large gap in the presence of albuminuria and assumption in the analysis, as shown in the validation analysis, in the younger population. (To begin with, could such prevalence of albuminuria be really true in the general young population?) And as the authors mentioned, although the testing costs may not be affected, several burdens such as psychological and cost will greatly affect this population. I feel it is important to be aware of any evidence on the prevalence of albuminuria in the young general population and its pathological significance. The relevance of this estimation analysis would differ according to the pathological and treatment related significance (with ACEI/ARBs) in this population. How do the authors relate to this? Would the evidence really suggest a conservative assumption for this model (eg if treating young people with ACEI/ARBs have no influence on future outcomes) ? Maybe to be more conservative, it could be suggested that this model be better use for those above 40 years old?

Our model utilizes data based on NHANES III, and if comparing the simulated with the published values, the difference is not big. The values published by Girndt et al. differ significantly from this, and we are unsure which values to rely on. Especially since a probably comparable population from the Netherlands in the PREVEND study provides data that match the values from NHANES III, but also Zhou et al., a more recent study from the US, provides even higher values than Girndt et al.

 18-19 20-29 30-39

Garg [1]

NHANES III 1988-1994 USA 5.7% (n=3071) 5.5% (n=2928)

Gracchi [2]

PREVEND 1997-1998 Groningen, NL ≈ 5.8% ≈ 5.7% (N=40000)

Girndt [3]

2008-2011 Germany 12.7% (n=205) 11.9% (n=781) 8.8% (n=830)

Zhou [4]

2011-2018 USA + Canada 24.7% 

(n =506)

We are not really able to solve this problem but decided to emphasize the importance of new research in this direction, as there is a considerable influence on our topic.

If we assume that these cases of “young albuminuria” have no impact on the incidence of renal replacement therapy in the later course, from a strict economic cost-effectiveness point of view, the influence from the younger age groups on the overall results is relatively small because the costs of an annual test (36,18€) are very similar to the costs of therapy for one year (54,74€) and due to the cohort-setting, 11% of all simulated person-years fall within the ages under 40. However, changing the period between the tests from one or two years increases this influence.

Until we are able to engage this topic with further insights, it is possible to limit the testing to individuals older than 40 to reach maximum conservativeness. We decided not to limit the testing for our exemplary results because we assess the S-GASM to be working as intended with the data it is based on, there is no final evaluation on the topic of albuminuria in younger individuals possible, and the changes in the results are to be estimated as minor.

We decided to add a section about this problem to the manuscript and also added this discussion to the supplementary material.

2) Line 163-166 the authors describe the factors integrated in the model but should not BMI be in there to?

This section (Line 159-163 in the revised manuscript) was meant to describe the parameters directly influencing the prevalence, incidence, and progression of albuminuria, the initial GFR, and annual GFR loss. BMI is influencing indirectly through higher risks for hypertension and diabetes but not directly. We added the word „directly“ to make this clearer.

Line 117-121 of the revised manuscript describe all factors integrated into the model.

1. Garg, A.X., et al., Albuminuria and renal insufficiency prevalence guides population screening: results from the NHANES III. Kidney Int, 2002. 61(6): p. 2165-75.

2. Gracchi, V., et al., Prevalence and distribution of (micro)albuminuria in toddlers. Nephrol Dial Transplant, 2016. 31(10): p. 1686-92.

3. Girndt, M., et al., The Prevalence of Renal Failure. Results from the German Health Interview and Examination Survey for Adults, 2008-2011 (DEGS1). Dtsch Arztebl Int, 2016. 113(6): p. 85-91.

4. Zhuo, M., et al., High Prevalence and Low Awareness of Albuminuria in the Community Setting in the KDSAP. Kidney Int Rep, 2020. 5(4): p. 475-484.

---

## [Decision Letter · Decision Letter 1]

20 Dec 2021

Development of the Simulation-based German albuminuria screening model (S-GASM) for estimating the cost-effectiveness of albuminuria screening in Germany

PONE-D-21-25470R1

Dear Dr. Kairys,

We’re pleased to inform you that your manuscript has been judged scientifically suitable for publication and will be formally accepted for publication once it meets all outstanding technical requirements.

Kind regards,

Tatsuo Shimosawa, M.D., Ph.D.

Academic Editor

PLOS ONE

Additional Editor Comments (optional):

Reviewers' comments:

Reviewer's Responses to Questions

**Comments to the Author**

1. If the authors have adequately addressed your comments raised in a previous round of review and you feel that this manuscript is now acceptable for publication, you may indicate that here to bypass the “Comments to the Author” section, enter your conflict of interest statement in the “Confidential to Editor” section, and submit your "Accept" recommendation.

Reviewer #1: All comments have been addressed

2. Is the manuscript technically sound, and do the data support the conclusions?

Reviewer #1: Yes

3. Has the statistical analysis been performed appropriately and rigorously? 

Reviewer #1: Yes

4. Have the authors made all data underlying the findings in their manuscript fully available?

Reviewer #1: Yes

5. Is the manuscript presented in an intelligible fashion and written in standard English?

Reviewer #1: Yes

6. Review Comments to the Author

Reviewer #1: All previous comments have been adequately addressed and I have no further comments to make.

7. PLOS authors have the option to publish the peer review history of their article (what does this mean?). If published, this will include your full peer review and any attached files.

Reviewer #1: **Yes: **Hiroo Kawarazaki

---

## [Editor Report · Acceptance letter]

27 Dec 2021

PONE-D-21-25470R1 

Development of the Simulation-based German albuminuria screening model (S-GASM) for estimating the cost-effectiveness of albuminuria screening in Germany 

Dear Dr. Kairys:

I'm pleased to inform you that your manuscript has been deemed suitable for publication in PLOS ONE. Congratulations! Your manuscript is now with our production department. 

Kind regards, 

on behalf of

Prof. Tatsuo Shimosawa 

Academic Editor

PLOS ONE